# Review of the Family Thanerocleridae (Coleoptera: Cleroidea) and the Description of *Thanerosus* gen. nov. from Cretaceous Amber Using Micro-CT Scanning

**DOI:** 10.3390/insects13050438

**Published:** 2022-05-06

**Authors:** David Peris, Bastian Mähler, Jiří Kolibáč

**Affiliations:** 1Departament de Dinàmica de la Terra i de l’Oceà, Facultat de Ciències de la Terra, Universitat de Barcelona, 08028 Barcelona, Spain; daperce@gmail.com; 2Institut de Recerca de la Biodiversitat (IRBio), Universitat de Barcelona, 08028 Barcelona, Spain; 3Institute of Geosciences, Section Paleontology, University of Bonn, 53115 Bonn, Germany; bastian.maehler@uni-bonn.de; 4Department of Entomology, Moravian Museum, 62700 Brno, Czech Republic

**Keywords:** computed tomography, Upper Cretaceous, Burmese amber, Thanerocleridae, Coleoptera

## Abstract

**Simple Summary:**

The use of new technologies applied to paleontological studies offers more detailed and attractive results each time. We describe a new genus and species of the relative rare beetle family Thanerocleridae (Cleroidea). *Thanerosus antiquus* gen. and sp. nov. is described as the fourth Cretaceous representative of Thanerocleridae. The set of characters observed in the new species suggests its basal position within the family and its relationship with the ancestral North American *Zenodosus sanguineus*, the single extant member of the subfamily Zenodosinae. The observance of some characters, including the mouthparts and details on the thorax’s ventral side, was only possible after the 3D reconstruction of the fossil following its X-ray micro-CT scanning. This fact reinforces the effectiveness of this non-destructive technique for evolutionary studies through the use of fossils. We speculate on a wide diversification and distribution of this predatory family originally connected with an ecological community of saproxylic insects in the Cretaceous that were followed by habitat change and specialization of life inside fruiting bodies of fungi. Consequently, the family has not been recorded from the Cenozoic fossil collections yet and currently shows limited diversification.

**Abstract:**

The predaceous beetle family Thanerocleridae is one of the smallest families of Cleroidea. It comprises only 36 extant species widespread on all continents. Three more species have been described from Cretaceous ambers of Myanmar and France. The fourth fossil representative of Thanerocleridae is described herein. *Thanerosus antiquus* gen. and sp. nov. is based on one fossil specimen preserved in an amber piece from Upper Cretaceous Kachin amber. The holotype was imaged using an X-ray micro-CT system to obtain high-quality 3D images. A phylogenetic analysis based on 33 morphological characters supports the placement of the new genus at the basal position in a tree of Thanerocleridae, in the vicinity of extant *Zenodosus* Wolcott and three extinct Mesozoic genera with which the new fossil shares open procoxal and mesocoxal cavities and transverse procoxae. We offer here a key to all extant and extinct genera in the family together with a complete list of all valid thaneroclerid taxa.

## 1. Introduction

Thanerocleridae is a small family of cleroid beetles comprising 36 living species in ten extant genera [1,2,3,4] and three fossil genera and species described from Cretaceous ambers [5,6,7]. One fossil described from the Yixian Formation (Lower Cretaceous), *Mathesius liaoningensis* Kolibáč and Huang (Cleroidea *incertae sedis*), was denoted a presumptive relative of the clerid or thaneroclerid branches of Cleroidea [8]. Still, the species has not been classified within any cleroid family yet.

The family is distributed worldwide, mostly in tropical and subtropical climates although one monotypic genus *Zenodosus* Wolcott lives in the temperate zone. Only the cosmopolitan single species *Thaneroclerus buquet* Lefebvre is known in Europe [1]. However, due to its Indian origin, it is not able to survive winter temperatures in the wild. It is likely that all thaneroclerid species are predatory. Their larvae and adults prey on small beetles associated with wood and fungi and can be collected in the same habitat as their prey, i.e., under the tree bark, beating the tree branches, or on fungi growing on trees [1]. Some species also show particular habitats, such as stored products or termite nests, where they prey, for example, on ptinids. However, the biology of the species is far from being completely known due to their relative scarcity [2].

Thanerocleridae constitutes one of the smallest families of the superfamily Cleroidea. It was initially included as a subfamily in Cleridae until Kolibáč [9] elevated it to a family rank. Kolibáč [10], in a review of cleroid classification, studied its phylogeny and divided the Cleroidea into melyrid, trogossitid, clerid and thaneroclerid branches. Thanerocleridae was maintained as a separate family by some authors [1,11,12], whereas it was re-classified within Cleridae by others [13,14]. Both suggestions recognized the thaneroclerid/-ine lineage as sister to the remaining lineages of Cleridae, but the authors chose alternate paths of family recognition without a defined position for some time [15]. Molecular data analyses have recovered Thanerocleridae in a separate family closely related to Cleridae, conforming to the clerid lineage [16,17,18], including Chaetosomatidae, Metaxinidae, Thanerocleridae and Cleridae [12,19]. Zhang et al. [16] suggested a divergence between Thanerocleridae and Cleridae in the Lower Cretaceous, about 110 million years (Ma). It was estimated a bit earlier but also in the Lower Cretaceous (about 135 Ma) by McKenna et al. [17], but occurred in the Lower to Middle Jurassic, according to Kolibáč et al. [19] and Cai et al. [18]. Such great divergence among estimations occurs when more fossil records of the superfamily are used for the calibration.

Currently, Thanerocleridae comprises two extant subfamilies, Zenodosinae and Thaneroclerinae, the latter with two tribes, Isoclerini and Thaneroclerini [9,11]. All extant members but one basal monotypic genus and species of the family (*Zenodosus sanguineus* (Say)) belong to the subfamily Thaneroclerinae [2,3,9,20,21,22]. Interestingly, all three fossil species described from the Cretaceous ambers have been classified within the recently monotypic Zenodosinae [5,6,7]. The extant *Z. sanguineus* occurs in southeastern Canada and the eastern, central, and southern United States [1]. The fossil distribution of the latter subfamily members in Europe and southeastern Asia in the Cretaceous ambers suggests that the range of the zenodosines was much wider in the Albian-Cenomanian than in the present [7].

*Thanerosus antiquus* gen. and sp. nov. is a further new thaneroclerid beetle described within Zenodosinae. The new genus is substantially different from the rest of the fossil representatives of the subfamily, providing evidence of a wider morphological variation of the group and showing an already demonstrated distribution of the lineage in the Cretaceous. It is the fourth species of Thanerocleridae to be described from the Mesozoic ambers, the third one from the same deposit of Kachin (Myanmar). However, its observation using a micro-CT scan suggests that this fossil might be a relative of the single extant zenodosine member *Zenodosus sanguineus* from North America more than the other burmite species.

## 2. Materials and Methods

### 2.1. Geological Setting

This study is based on one fossil specimen found in a sample of Kachin amber (Myanmar). The amber from Myanmar, many times referred to as Burmese amber, has a long history of excavation. The active amber mines in Myanmar have increased in the last years [23,24]. This fact forced the use of more defining terms to refer to the general Burmese or Myanmar amber. The Kachin amber is commonly cited as being from an area near Noije Bum peak, around 20 km to the southwest of Danai (=Tanai) in Hukawng Valley, Kachin Province, northern Myanmar (Figure 1). However, another less known area is also 35 km to the southeast of Hkamti [23,25]. The amber is preserved within Cretaceous flysch-type units. The surrounding matrix consists of fine-grained sedimentary rock, greyish to bluish-green in color, with fine fragments of volcaniclastic sediments [26]. These amber deposits are lower Cenomanian in age (98.79 ± 0.62 Ma), according to current dating using U-Pb zircon crystals obtained from the amber matrix [26]. There exists a current discussion about the true age of this amber, which could be slightly older due to the high roundness of the amber surface and the presence of marine faunal inclusions, which indicates a reworking process before the amber was buried in the surrounding rock matrix. Some other amber sources in Myanmar have recently been dated from ~110 to ~72 Ma [23,24]. As such, the refined understanding of amber deposits in Myanmar highlights the importance of distinguishing sources.

### 2.2. Methods of Observation

The holotype denoted as NIGP180154 is included in a polished oval sample of amber along with one syninclusion of a Diptera specimen (probably Tipulidae). The sample was further ground and polished to better observe the characters of the specimens. It was examined under a Leica MZ95 stereomicroscope (Leica Microsystems, Heerbrugg, Switzerland) and a Leica DME compound microscope (Leica Microsystems, Heerbrugg, Switzerland). Detailed photomicrographs of the fossil were created using an Olympus CX41 compound microscope (Olympus, Tokyo, Japan) and a Leica MS5 stereomicroscope (company Leica Microsystems, Heerbrugg, Switzerland), both equipped with a digital camera sCMEX-20 (Euromex Microscopen, Arnhem, The Netherlands), under incident light and using the software ImageFocusAlpha version 1.3.7.12967.20180920 (Euromex Microscopen, Arnhem, The Netherlands), finally merged with the software CombineZP.

The holotype was imaged at the Nanjing Institute of Geology and Palaeontology (NIGP) with a Zeiss Xradia 520Versa X-ray micro-CT system to obtain high-quality 3D images. Considering the comparatively small size of the fossil, a CCD-based 4× objective was used, providing isotropic voxel sizes of 2.3938 μm with the help of geometric magnification. Images were generated at an X-ray voltage of 50 kV. Four frames per projection were acquired with an integration time of 3.5 s for 2801 projections over 360°. Acquired images were rendered and visualized using Avizo 8.1 (Thermo Fisher Scientific, Schwerte, Germany). Video presentations are included in Appendix A.

The final images were edited with Adobe CS6. All relevant structures were measured from the digitized images. The specific terminology for characters follows that of Kolibáč & Leschen [1]. We followed the family-group classification of Gimmel et al. [12]. The nomenclatural acts established herein are registered under ZooBank LSID: 9BC3609F-64F6-4AAB-AC8C-7683508BF73E.

### 2.3. Phylogenetic Analysis

An analysis of 33 morphological characters was used to find an approximate position of the fossil within Thanerocleridae and reveal the extant representatives’ phylogeny. The following extant species of Cleroidea were used for a character analysis as outgroups: *Acanthocnemus nigricans* Hope (Acanthocnemidae), *Tenebroides mauritanicus* Linnaeus (Trogossitidae), *Tilloidea transversalis* Charpentier (Cleridae: Tillinae), and *Clerus mutillarius* Fabricius (Cleridae: Clerinae). Trees were rooted with *Acanthocnemus nigricans,* which was revealed as the basal taxon in the specialized molecular phylogeny of Cleroidea by Gimmel et al. [12]. The data matrix of morphological characters was assembled in WinClada 1.00.08 [27]. TNT 1.5 [28,29] was used for parsimony analysis using the implicit enumeration strategy. The characters were unordered, and all were switched as non-additive (see Appendix A for a list of the characters and the matrix). The unknown character states were denoted by the question mark (?), the lacking characters by the dash (–). Two analyses were conducted: (1) with character state 7 (1) (mandible bidentate) exactly as mentioned in particular descriptions of *Mesozenodosus insularis* Tihelka et al., 2020 and *Cretozenodosus fossilis* Cai & Huang, 2018; (2) with modified character state 7 (0) (mandible unidentate) and state unknown 7 (?), respectively. Parsimony analysis was conducted with implied weighting (K = 12; see Goloboff et al. [30] and Smith [31]). Tree support was measured as Bremer support (TBR, 1000 steps) conducted in TNT 1.5.

## 3. Results

### Systematic Palaeontology

Order Coleoptera Linnaeus, 1758

Suborder Polyphaga Emery, 1886

Superfamily Cleroidea Latreille, 1802

Family Thanerocleridae Chapin, 1924

Subfamily Zenodosinae Kolibáč, 1992


***Thanerosus* gen. nov. Peris & Kolibáč**


Figure 2, Figure 3 and Figure 4

*Type species*. *Thanerosus antiquus* gen. and sp. nov. here designated.

*Etymology*. The generic name is derived from the prefix ‘*Thanero*-’ following the family name, and ‘-*sus*’, termination of the most basal extant representative of the family (*Zenodosus* Wolcott). It is masculine in gender. The genus is registered under Zoo-Bank LSID: 84DA2BE8-C471-44F9-A4AF-FEE0C6EE6F2E.

*Type*, *locality*. Kachin State, near Tanai, northern Myanmar; Upper Cretaceous (lower Cenomanian) in age [26].

*Diagnosis*. Dorsal surface with dense and elongate setae; terminal palpomere of maxillary palps coniform; labial palpi with three segments, terminal palpomere weakly securiform; 11-segmented antenna with loose but distinct 3-segmented club; anterior edge of clypeus straight, labrum distinctly emarginate medially; mandible unidentate, incisor edge with single blunt subapical teeth; prothorax with lateral carina visible along the whole length; procoxal cavities externally open, procoxae weakly transverse, pair of depressions along notosternal sutures present; scutellum quadrate; all tibiae with two spurs present at apex; tarsal formula 5-5-5, meso and metatarsomeres 2–4 with conspicuous lobes, empodium bisetose.

*Remarks*. *Thanerosus* gen. nov. can be referred to Thanerocleridae based on the general body shape, head prognathous and its base as wide as pronotum; prothorax with complete lateral carina, the base of pronotum bordered; all coxae narrowly separated; metacoxae extending laterally to meet elytra, metanepisternum elongate; protarsomeres 1–4 wide with tarsi compact, tarsal claws simple; abdomen with five ventrites [5,9,13]. The family has some other apomorphic characters that are difficult to observe in fossil specimens, even after 3D reconstruction, such as tegmen without the median strut and lateral struts or four malpighian glands [9].

The recent genus *Zenodosus* and all three fossil thaneroclerids described to date are placed in Zenodosinae based on the procoxal cavities open and the procoxae transverse. By contrast, species in Thaneroclerinae possess the procoxal cavities externally closed and procoxae more or less spherical [9]. The shape of both maxillary and labial terminal palpomeres in *Thanerosus* gen. nov. also corresponds with zenodosines, whereas depressions along notosternal are known only in the tribe Isoclerini.


***Thanerosus antiquus* gen. and sp. nov. Peris & Kolibáč**


Figure 2, Figure 3 and Figure 4

*Holotype*. Holotype NIGP180154, adult specimens, sex unknown. The type specimen is deposited in the Nanjing Institute of Geology and Palaeontology, Chinese Academy of Sciences, Nanjing, China. The holotype lost the apical section of the left antenna before being embedded in resin.

*Type locality*. Kachin State, near Tanai, northern Myanmar; Upper Cretaceous (lower Cenomanian) in age [25].

*Etymology*. The specific epithet is designated after the Latin *antiquus*, meaning old or ancient. It is masculine in gender. The species is registered under Zoo-Bank LSID: 13AB22E2-8519-4C5A-BF09-0926A7889884.

*Diagnosis*. As for the genus.

*Description*. Body length dorsally 2.24 mm; length of elytron 1.41 mm, the width of elytron at base 0.85 mm, a maximum width of pronotum 0.7 mm, length of pronotum 0.66 mm. Body 1.59 times as long as wide.

Coloration and structure: body elongate and subcylindrical (Figure 2A); winged; generally black and dark brown (Figure 4A). The body surface is densely clothed with long, erect hairs, more conspicuous dorsally. Body densely punctate, interspaces smaller than the diameter of punctures in the head and pro- and mesosternum, interspaces larger than the diameter of punctures in pronotum and elytra. Metaventrite, abdominal ventrites and legs without sculpture. Elytral surface with punctation regular but disordered. Pubescence of ventral surface scarce, conspicuous only in legs.

Head: prognathous, as wide as prothorax, artificially declined in fossil; sharply incised epicranial acumination present (Figure 2A); frontoclypeal suture absent, anterior edge of clypeus straight; labrum distinctly emarginate (Figure 2B). Gular sutures widely separated and convergent. Eye relatively large and protuberant, without interfacetal setae, anteriorly with minute emargination. Antennal insertions are laterally situated and weakly concealed from above. Mandible robust, broad basally, abruptly curved, with single apical tooth; incisor edge with single blunt subapical teeth. Maxillary palpi 3-segmented, first and last palpomeres equal in length, second one about 0.3 times of length of third palpomere; terminal palpomere coniform (Figure 3B,C). Ligula deeply emarginate; labial palpi long, projecting forward between mandibles, 3-segmented, terminal palpomere securiform (Figure 3C,D). Antenna longer than to base of elytron, extending slightly backward, with eleven antennomeres, moniliform, with a loose 3-segmented apical club (Figure 3C); scape is enlarged and slightly wider than pedicel, antennomeres 3–8 subequal in width with antennomere 8 wider than long, antennomeres 9–11 abruptly widened and twice wider than segment 8; antennomeres 9–10 widest apically, antennomere 11 apically rounded.

Thorax: pronotum subquadrate, about as long as wide, convex, widest at one-third from the base and only very slightly narrowed towards base, lateral sides very weakly arcuate; pronotum without depressions or grooves, convex. Lateral pronotal carina complete; the base of the pronotum carinate (Figure 2E); anterior and posterior angles rounded. Depression along notosternal suture present (Figure 2C–E). Prosternum in front of coxa as long as coxal diameter; prosternal process about 0.3 times width of coxa, parallel-sided, extending beyond the posterior margin of procoxae. Procoxal cavities oval, open externally. Procoxa transverse and oval, narrowly separated, protrochantin exposed (Figure 2C). Scutellum quadrate, with apical margin arrowhead-like. Mesocoxal cavities are circular, narrowly separated by about 0.2 times their width. The subtriangular anterior process of the metaventrite extending midway between mesocoxae. Metaventral discrimen present; metanepisternum distinct, elongate, narrower posteriorly. Metacoxae flat, transverse, very narrowly separated medially, extending laterally to meet elytra. Elytron elongate, 1.6 times longer than wide, 1.2 times wider than the pronotum; elytra parallel-sided, without carinae; humeral angles not protuberant, smoothly rounded; epipleuron complete, widest basally and narrowing posteriorly.

Legs: slender and long, setose. Trochanters triangular. Femora widest near middle, weakly grooved for the reception of tibiae. Tibiae is widest apically, each with two short apical spurs (Figure 4B,D). Tarsi 5-5-5 (Figure 4B–D); protarsomeres 1–4 strongly expanded, with distinct ventral lobes, tarsus compact (Figure 2E and Figure 4B); tarsomeres 2–4 in remaining pairs of legs weakly expanded, with short ventral lobes (Figure 4C,D). Last tarsomere in all pairs of legs as long as preceding four segments together. Tarsal claws are simple, without claws; empodium well developed and bisetose, moderately projecting (Figure 4C).

Abdomen: five smooth ventrites present, without visible punctation and pubescence (Figure 2C); first ventrite 1.6 times longer than second; ventrite 2 slightly longer than 3; ventrites 3–5 of the same length; terminal ventrite (sternite VII) rounded at apex.

## 4. Discussion and Conclusions

### 4.1. Classification of Thanerosus antiquus gen. and sp. nov.

Thanerocleridae had been originally classified within the Cleridae until Kolibáč [9] raised the taxon to family rank, supporting the monophyly of the family in a list of characters that differ from Cleridae, most notably by having the minutely emarginate eyes, expanded protarsomeres 1–4, mostly five visible abdominal ventrites and aedeagus without median and lateral struts. *Thanerosus antiquus* gen. and sp. nov. are placed in Zenodosinae, as the other three fossil species of the family, based on the procoxal cavities open and procoxae transverse (Figure 2C). The new genus differs from the other thanerocleride genera in the scutellum quadrate. It differs from *Zenodosus* in the loose but distinct antennal club and regular elytral sculpture, while the club is weak and sculpture irregular in *Zenodosus* [9]. *Mesozenodosus insularis* Tihelka, Cheng, Huang, Perrichot & Cai, the most ancient fossil of the family from French amber [7], has the eyes coarsely setose with interfacetal setae, distinct depression in apical antennomere, pronotom with lateral carina weakly developed, the base of pronotum narrowed, depression along notosternal suture absent and scutellum strongly transverse. According to the original description, the mandibles are bidentate in *Mesozenodosus,* but such a feature is unknown in the all clerid lineage and the figure of the mandible by Tihelka et al. [7] (p. 393, Figure 3C) shows the unidentate apex of the mandible. By contrast, *Thanerosus antiquus* gen. and sp. nov. has the eye glabrous, depression in the apical antennomere absent, lateral pronotal carina distinct, pronotum only slightly constricted towards the base, depression along notosternal suture present and scutellum quadrate (Figure 2). The new species is also different from both previously described species from the Kachin amber. *Archaeozenodosus bellus* Yu & Kolibáč is more than two times bigger, clothed with short setae, pronotum widened anteriorly, depression along notosternal suture absent, procoxal cavity oval (but not spherical), scutellum transverse and broadly rounded apically, and elytral base about as broad as the base of pronotum [5]; *Thanerosus antiquus* gen. and sp. nov. is much smaller in length, clothed dorsally with long setae, pronotum widest at one-third from its base, depression along notosternal suture present, procoxal cavity transverse, scutellum quadrate and apically arrowhead-shaped, and elytral base distinctly wider than the base of pronotum. *Cretozenodosus fossilis* Cai & Huang lacks depression along notosternal suture and base of pronotum half the width of elytral bases [6] while depression along notosternal suture is present and the base of the pronotum is only slightly narrower than elytral bases in *Thanerosus antiquus* gen. and sp. nov. Moreover, the description of *Cretozenodosus* mentions bidentate mandibles; however, this structure is not figured and is highly improbable in Thanerocleridae.

### 4.2. Phylogeny and Palaeontology of Thanerocleridae

Parsimony analyses yielded in (1) the single most parsimonious tree (L = 58, Ci = 67, Ri = 78) (Figure 5A) and (2) three trees, the strict consensus of which (L = 59, Ci = 66, Ri = 77) is shown in Figure 5B.

Sister relation of the family Thanerocleridae and Cleridae is well-supported in our analysis (Bremer support > 1), as similarly occurred in previous studies [12]. Kolibáč [9] proposed *Zenodosus* as the most ancestral member of the family. A proof supporting the idea of Zenodosinae as the basal thaneroclerid group is that all the fossils described in the family so far, always from Cretaceous ambers, are different genera of the same subfamily. All four fossil species possess ancestral characters such as externally open procoxal cavities (12-0) and transverse or oval procoxae (13-0).

*Zenodosus* and *Thanerosus* share the open mesocoxal cavities (14-0) (mesepimeron touches mesocoxa) while all other thaneroclerids but *Ababa* Casey (character unknown in *Mesozenodosus*) have the cavities closed by projections of meso- and metaventrite which character state is considered apomorphic (14-1). That is the reason for the sister relation between *Zenodosus* and *Thanerosus* shown in Figure 4A. However, the analysis with unresolved relations among five ‘zenodosine’ taxa (Figure 4B) is more realistic because the relations figured in the previous analysis are, excepting mesocoxal cavities open/closed, based on unknown character states (?) in the fossil taxa and dubious observation of bidentate mandible in *Cretozenodosus* and *Mesozenodosus* (see above). No single taxon in the whole clerid lineage (Chaetosomatidae, former Metaxinidae, Thanerocleridae, Cleridae) has the bidentate mandible with two apical teeth situated side by side (or in the horizontal axis) as in other cleroids and the major part of cucujoid beetles. Moreover, as already noted above, the figure of the *Mesozenodosus* mandible shows the unidentate mandible [7].

The subfamily Thaneroclerinae is supported with three apomorphies: procoxal cavities externally closed (12-1/2), procoxa almost spherical (13-1), wing with radial cell absent (25-1). The clade *Ababa* + (*Isoclerus* Lewis, *Compactoclerus* Pic, *Parathaneroclerus* Pic) (the tribe Isoclerini) is based on a flat eye not exceeding the contour of the head (4-0) and especially unique synapomorphy 18-1 (tarsal formula 5-4-4). The clade comprising the five remaining extant genera *Onerunka* Kolibáč + (*Thaneroclerus* Lefebvre, *Neoclerus* Lewis + (*Meprinogenus* Kolibáč + *Viticlerus* Miyatake)) (the tribe Thaneroclerini) shares apomorphic irregular sculpture of elytron (23-1).

Up to now, Melyridae, Mauroniscidae, Prionoceridae, Lophocateridae, Trogossitidae and Cleridae are the six cleroid families found in the Middle Jurassic of northeastern China [32]. Fossil descriptions and the calibrated molecular clock suggested that the clerid lineage was fully developed and well-differentiated from other cleroid lineages in the middle Jurassic [15,19]. In the molecular analyses, Gimmel et al. [12] showed Thanerocleridae as a sister to Cleridae, while Kolibáč et al. [19] considered it a sister to Chaetosomatidae and estimated the split event of both clades using Bayesian tip dating analysis to the Lower Jurassic. Finally, Cai et al. [18] estimated the Cleridae-Thanerocleridae split approximately in the mid-Jurassic (Chaetosomatidae was not included in the analysis). Although Jurassic thaneroclerid fossils are unknown, we already know that Thanerocleridae was well diversified and distributed by the mid-Cretaceous. Strange enough, the family has not been recorded from the big collections of Cenozoic ambers, revealing a possible loss of diversity, paleogeographic constrictions or different natural history of the thaneroclerids since the Upper Cretaceous. The latter reason may be related to the specialization of the thaneroclerids on life in fungi. At the same time, ancestral members of the clerid lineage are predatory and live under bark or in galleries of wood-boring insects similar to *Zenodosus sanguineus*, the major part of extant thaneroclerids (the species-richest genera *Isoclerus* and *Neoclerus*) hunt for prey in a tree or freely growing fungi. Probably, the four extinct thaneroclerids, including *Thanerosus antiquus* gen. and sp. nov., found in European and Asian Cretaceous ambers, had the ancestral biological pattern connected with the ecological community of saproxylic insects, like many other groups of amber bearing beetles from the Cretaceous ambers [33]. It is possible that Cretaceous thaneroclerids were living under bark or on logs where the likelihood of sinking into the resin was higher than for species dwelling in fungal fruiting bodies [34]. Although characters in the cleroid mouthparts together with molecular studies suggest that primitive cleroids were mainly fungus-feeding [12,19], as similarly suggested for other groups in Coleoptera [35], and predatory shifts and flower-feeding in Cleroidea occurred presumably later, predation on insects living in fungi is probably derived in the clerid lineage from common hunting on bark surface or inside galleries of wood borer larvae as we can observe in the ancestral *Zenodosus sanguineus*.

### 4.3. Key to the Extinct and Extant Genera of Thanerocleridae

Kolibáč [9] proposed a key to classify the higher taxa in the family. An improved version of the key was needed after the recent description of additional extant and extinct taxa in the family. The genus *Cleridopsis* Champion (type of genus: *C. latimanus* Champion by monotypy) from Central America (Guatemala, Panama) was described within Cryptophagidae [36] (pp. 60, 94–95); however, the original description and illustrations [36] (pp. 94–95, Pl. III: Figures 10, 10a) perfectly determine its familial membership. Corporaal [37] had not noted this fact, but Crowson [38] (p. 310) recognized it and rightly affiliated *Cleridopsis* with Thanerocleridae. Kolibáč [9] (p. 338, footnote) did not include the genus in his family revision because he had not studied the two only known *C. latimanus* specimens. A synonymization of *Cleridopsis* with *Ababa* from the same geographic region has been recently made by Opitz [22], who also restituted the generic rank for *Ababa* and *Parathaneroclerus,* although the taxa are extremely similar and rather congeneric. Therefore, *Cleridopsis* is not included in the following key, and both latter taxa are treated as genera.

It should also be noted here that *Allothaneroclerus* Corporaal actually possesses the formula 5-4-4, not 5-5-5, as erroneously observed by Opitz [22]. Moreover, the latter genus has already been synonymized with *Isoclerus* by Kolibáč [9]. Therefore, *Isoclerus tuberculatus* Schenkling (originally *Neoclerus*) and the related *I. succedaneus* Melnik belong to Isoclerini and not to Thaneroclerini as suggested by Opitz [22] (p. 16).


1.Procoxal cavities externally widely open, procoxae oval or transverse; abdominal segment 9 fully developed; discriminal line present; tarsal formula 5-5-5; mesocoxal cavities externally open or closed. **Subfamily Zenodosinae**.................................**2**
–Procoxal cavities externally perfectly, rarely imperfectly (*Ababa*, *Parathaneroclerus*) closed; procoxa more or less spherical; abdominal segment 9 reduced to spicular fork (tergite and sternite VIII inconspicuous); discriminal line scarcely perceptible; tarsal formula 5-5-5 or 5-4-4; mesocoxal cavities externally closed, rarely open (*Ababa*). **Subfamily Thaneroclerinae**........................................................................**6**
2.Mesocoxal cavities externally open...............................................................**3**
–Mesocoxal cavities externally closed; if the latter feature unknown, a body under 2.5 mm; terminal antennomere with concavity......................................................**4**
3.Antennal club weak and indistinct; scutellum transverse; North America.........................................................................................................***Zenodosus* Wolcott**
–Antennae with a loose but distinct club; scutellum quadrate................................................................................................**† *Thanerosus* gen. nov. Peris & Kolibáč**
4.Eyes without interfacetal setae; pronotal carina well-developed; terminal antennomere without concavity; body > 2.5 mm...............................................**5**
–Eyes with interfacetal setae; pronotal carina weakly developed; terminal antennomere with concavity; body size < 2.5 mm.................................................................................† ***Mesozenodosus* Tihelka, Cheng, Huang, Perrichot & Cai**
5.Pronotum widest anteriorly; procoxal cavity oval; elytral bases about as broad as the base of pronotum; more than 5 mm in length.......**† *Archaeozenodosus* Yu & Kolibáč**
–Pronotum widest at 1/3 from the base; procoxal cavity transverse; the base of pronotum half the width of elytral bases; less than 5 mm in length.............................................................................................**† *Cretozenodosus* Cai & Huang**
6.Tarsal formula 5-5-5; depression along notosternal suture absent; procoxae subspherical and moderately narrowly separated. **Tribe Thaneroclerini**...............**7**
–Tarsal formula 5-4-4, depression along notosternal suture present; procoxae spherical and extremely narrowly separated. **Tribe Isoclerini**.......................................**11**
7.Tarsomere 4 of protarsi distinctly smaller than 3; pronotal depressions present; unicolorous black or brown species...............................................................**8**
–Tarsomeres 2, 3 and 4 of protarsi subequal in size; pronotal depressions are absent......................................................................................................**9**
8.Radial cell present; elytra without tufts of hairs; mostly five abdominal ventrites; southeastern and eastern Asia, cosmopolitan..................***Thaneroclerus* Lefebvre**
–Radial cell absent; each elytron with a humeral tuft of black hairs and three tufts of white, stout, semierect or decumbent hairs; six abdominal ventrites; India........................................................................***Meprinogenus* Kolibáč**
9.Pronotum laterally rounded; protibia without any apical spine at the apex.........**10**
–Pronotum parallel-sided; protibia with one short, blunt, not hooked apical spine at apex; New Guinea...............................................................***Onerunka* Kolibáč**
10.Antennomeres 10 and 11 coalescent (suture perceptible); elytron not depressed, without distinct humeral gibbae; abdominal ventrite 1 as long as 2 and 3 together, 2 to 5 subequal in length; winged species; southeastern and eastern Asia...........................................................................................................***Neoclerus* Lewis**
–Antenna 11-segmented, club loose; elytron strongly depressed in anterior half, with elevated humeral gibbae; abdominal ventrite 1 as long as 2 to 4 together, 2 to 6 successively shorter; single wingless species; Fiji.....................***Viticlerus* Miyatake**
11.Tendency to a coalescence of antennomeres 10 and 11 (suture between them perceptible), club 2-segmented; procoxae spherical, very small, space between them minute; prosternum in front of procoxa very long; terminal palpomere of labial palpi coniform; Africa, Madagascar.............................................***Compactoclerus* Pic**
–Antennae 11-segmented with a more or less distinct 3-segmented club; terminal palpomere of labial palpi truncate; Americas, southeastern Asia........................**12**
12.Procoxal cavities externally perfectly closed; southeastern Asia......***Isoclerus* Lewis**
–Procoxal cavities externally imperfectly closed; Americas.................................**13**
13.Mesocoxal cavities open; pronotum with weak depressions or flat; Americas......................................................................................................***Ababa* Casey**
–Mesocoxal cavities closed; pronotum with three conspicuous depressions; Brazil.........................................................................................................**Parathaneroclerus Pic**


### 4.4. Checklist of the Family Thanerocleridae Chapin, 1924


**Subfamily Zenodosinae Kolibáč, 1992**


Genus † *Archaeozenodosus* Yu & Kolibáč, 2017 (type species: monotypic)

† *bellus* Yu & Kolibáč, 2017 Burmese amber: Cenomanian

Genus † *Cretozenodosus* Cai & Huang, 2018 (type species: monotypic)

† *fossilis* Cai & Huang, 2018 Burmese amber: Cenomanian

Genus † *Mesozenodosus* Tihelka, Cheng, Huang, Perrichot & Cai, 2020 (type species: monotypic)

† *insularis* Tihelka, Cheng, Huang, Perrichot & Cai, 2020 Charentese amber: Cenomanian

Genus † *Thanerosus* gen. nov. Peris & Kolibáč (type species: monotypic)

† *antiquus* sp. nov. Peris & Kolibáč Burmese amber: Cenomanian

Genus *Zenodosus* Wolcott, 1910 (type species: monotypic)

*sanguineus* Say, 1835 (*Clerus*) Canada, USA


**Subfamily Thaneroclerinae Chapin, 1924**



**Tribe Thaneroclerini Chapin, 1924**


Genus *Meprinogenus* Kolibáč, 1992 (type species: monotypic)

*indicus* Corporaal, 1939 (*Cyrtinoclerus*) India: Tamil Nadu

Genus *Neoclerus* Lewis, 1892 (type species: *Neoclerus ornatulus* Lewis, 1892)

*nanus* Schenkling, 1901 (*Thaneroclerus*) Indonesia: Java, Borneo, Sumatra; Malaysia: Malacca

*nilgiriensis* Corporaal, 1939 India: Tamil Nadu

*notatus* Pic, 1930 “Tonkin” (North Vietnam/Laos)

*ornatulus* Lewis, 1892 Japan, Taiwan

quinquemaculatus Gorham, 1892 (Thaneroclerus) Myanmar

Genus *Onerunka* Kolibáč, 2012 (type species: monotypic)

*longi* Kolibáč, 2012 Papua New Guinea

Genus *Thaneroclerus* Lefebvre, 1838 (type species: *Clerus buquet* Lefebvre, 1838)

*buquet* Lefebvre, 1838 (*Clerus*) cosmopolitan, origin probably in India

*impressus* Pic, 1926 “Tonkin” (North Vietnam/Laos)

*ishigakiensis* Murakami, 2016 Japan

*quasitardatus* Corporaal, 1939 India: Kashmir

*termitincola* Corporaal, 1939 Indonesia: Sumatra

Genus *Viticlerus* Miyatake, 1977 (type species: monotypic)

*formicinus* Miyatake, 1977 Fiji: Viti Levu


**Tribe Isoclerini Kolibáč, 1992**


Genus *Ababa* Casey, 1897 (type species: *Clerus tantillus* Le Conte, 1865 = *Ababa crinita* Casey, 1897)

*adona* Opitz, 2018 Costa Rica, Panama

*epiiska* Opitz, 2018 Argentina, Bolivia, Peru

*granaria* Opitz, 2018 Mexico

*latimana* Champion, 1913 (*Cleridopsis*) Guatemala, Panama

*tantilla* Le Conte, 1865 (*Clerus*) USA: Alabama, Illinois, Florida, Texas, Washington DC: Mexico; Panama

Genus *Compactoclerus* Pic, 1939 (type species: *Compactoclerus robustus* Pic, 1939)

*davidi* Kolibáč, 1992 Congo

*robustus* Pic, 1939 Congo

*sicardi* Pic, 1939 (*Microababa*) Madagascar

*zambiensis* Kolibáč, 1999 Zambia

Genus *Isoclerus* Lewis, 1892 (type species: *Isoclerus pictus* Lewis, 1892)

*cipisek* Kolibáč, 1998 Australia: New South Wales

*disinlei* Kolibáč, 1992 Taiwan

*elongatus* Schenkling, 1906 (*Thaneroclerus*) China: Yunnan

*gerstmeieri* Kolibáč, 1998 Australia: New South Wales, Queensland

*manka* Kolibáč, 1998 Australia: New South Wales

*menieri* Kolibáč, 1992 Indonesia: Lombok

*parallelus* Lewis, 1892 (*Lyctosoma*) China: Sichuan; Japan; Indonesia: Java, Sumatra; Vietnam

*pictus* Lewis, 1892 Japan

*rumcajs* Kolibáč, 1998 Australia: Queensland

*sarawacensis* Corporaal, 1939 Malaysia: Sarawak

*succedaneus* Melnik, 2005 India: Tamil Nadu

*tuberculatus* Schenkling, 1906 (*Neoclerus*) Sri Lanka

Genus *Parathaneroclerus* Pic, 1936 (type species: monotypic)

*triimpressus* Pic, 1936 Brazil: Rio Grande do Sul

## Figures and Tables

**Figure 1 insects-13-00438-f001:**
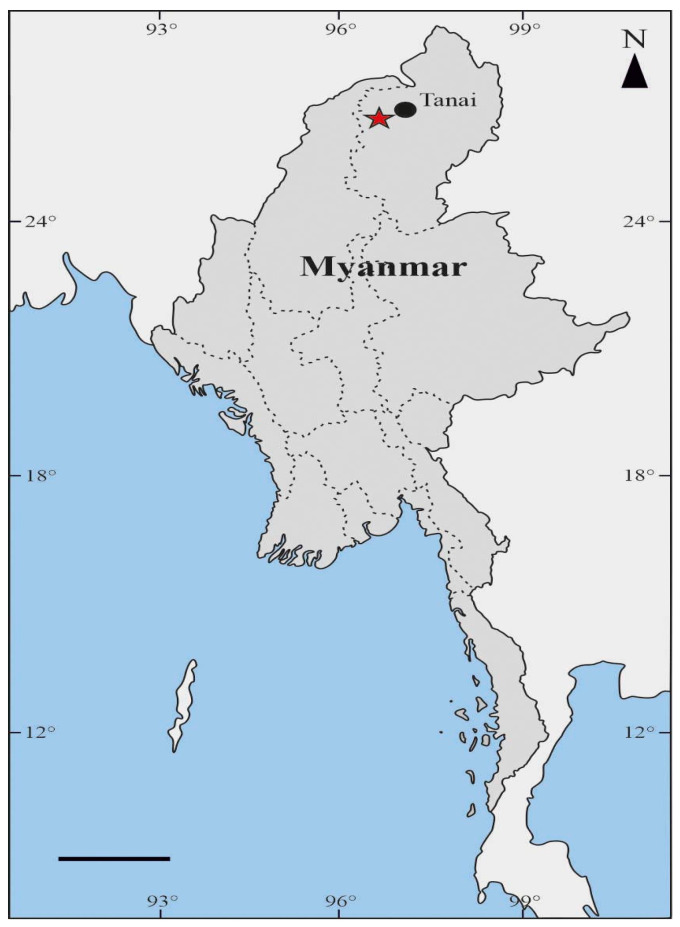
Map of Myanmar. The star marks the location of the Noije Bum hill mines in the Hukawng Valley, Kachin state, 18 km southwest of the town of Tanai (northern Myanmar). Scale bar 200 km.

**Figure 2 insects-13-00438-f002:**
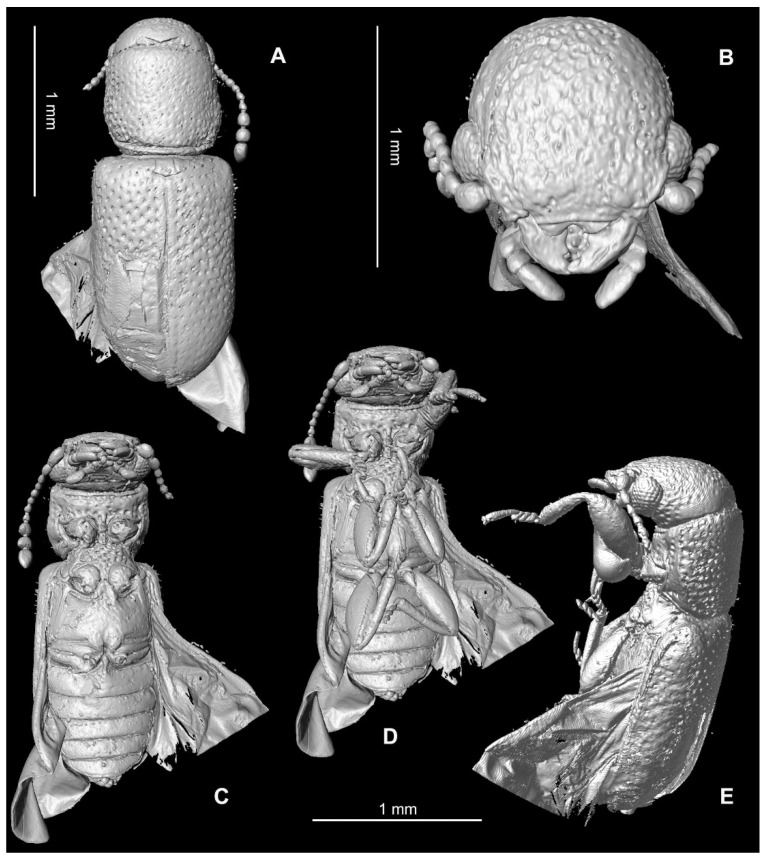
μCT reconstruction of *Thanerosus antiquus* gen. and sp. nov. holotype NIGP180154: (**A**) body, dorsal view; (**B**) head, anterior view; (**C**) body, ventral view; legs removed; (**D**) body, ventral view including legs; (**E**) body, lateral view.

**Figure 3 insects-13-00438-f003:**
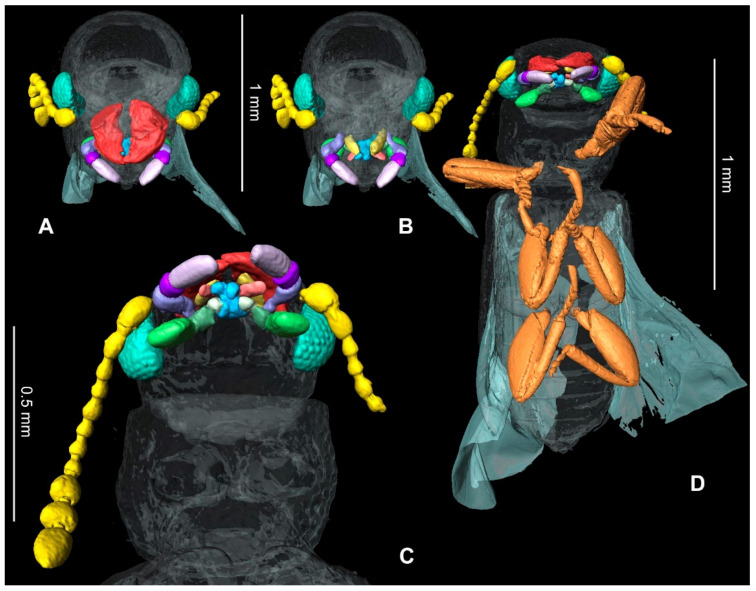
Detailed reconstruction of the eyes, antennae, mouthparts, and legs: (**A**) head anteriorly, including mandibles; (**B**) head anteriorly excluding mandibles; (**C**) head ventral view; (**D**) body ventrally with highlighted eyes, antennae, mouthparts and legs.

**Figure 4 insects-13-00438-f004:**
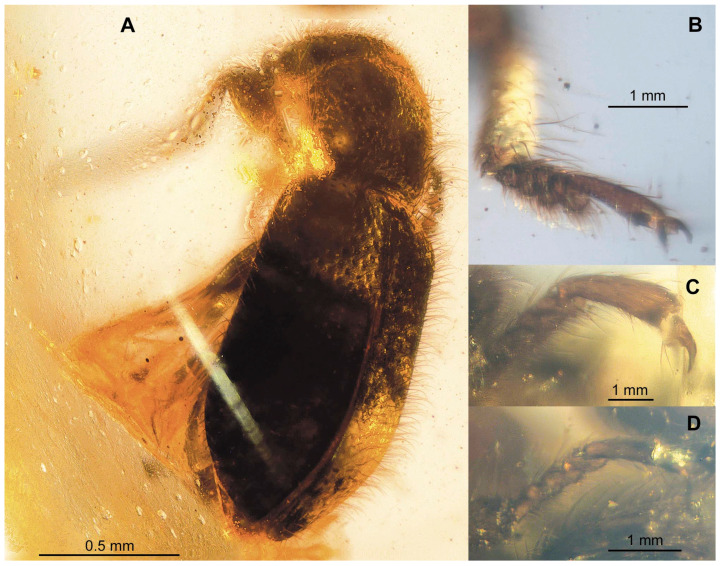
Pictures of *Thanerosus antiquus* gen. and sp. nov. holotype NIGP180154: (**A**) body of the holotype in a curved amber surface; (**B**) protarsi; (**C**) mesotarsi; (**D**) metatarsi.

**Figure 5 insects-13-00438-f005:**
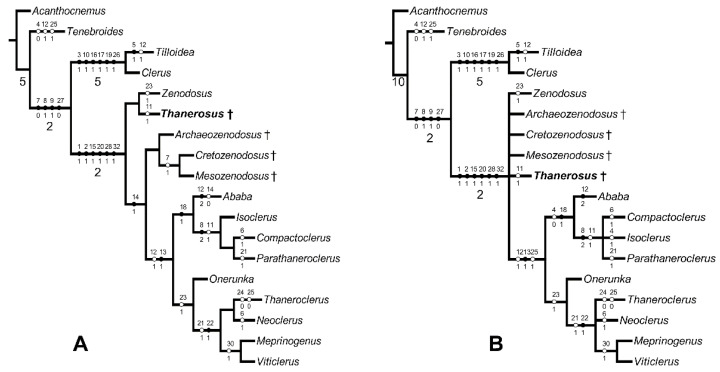
Phylogenetic trees with mapping of character states and Bremer support values above 1 (below lines): (**A**) single tree resulting from the analysis in which character 7 was treated as in original descriptions (mandible bidentate in *Mesozenodosus* and *Cretozenodosus*); (**B**) strict consensus of three trees with character 7-0 (mandible unidentate) in all fossils but *Cretozenodosus* in which character 7 is denoted as unknown (7-?).

## Data Availability

The data presented in this study are available in the article and in the Appendix A.

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
