# Peer review of "Review of the Family Thanerocleridae (Coleoptera: Cleroidea) and the Description of Thanerosus gen. nov. from Cretaceous Amber Using Micro-CT Scanning"

_insects, 2022, doi:10.3390/insects13050438_

Round 1

Reviewer 1 Report

Overall a well presented and scientifically rigorous presentation of a basal member of Cleroidea. The CT scans are outstanding and are a good example of the use of this new revolutionary method of interpreting morphological details in amber specimens. The taxonomic results will likely be of interest to relatively few specialists, but are important within that group. The methodology will be of much broader interest. 

The English presentation needs to be improved. There are some common awkward word choice and a few grammatical mistakes. These are easily corrected but likely will require the assistance of a technically proficient English first language editor/reviewer. 

Author Response

Thank you for the positive review. We corrected some technical and formal issues (the holotype information, LSID), formal arrangement of the Key and some minor mistakes. The manuscript was proof-read by a native speaker to work up its English.

Reviewer 2 Report

The article presented have high significance in our knowledge of extinct beetle species and presents new data on Cleroidea fossils. I  am pleased recommend to publish this paper as it is presented. 

The manuscript proposed is devoted to a new method study of the Cretaceous amber inclusion resulted description of a new genus and species of the cleroid family Thanerocleridae. Both new method of amber inclusion study and presentation of ancient beetle taxa present original topic of paleontological studies of invertebrates. Each fossil biota evidence that could be studied and implemented to phylogenetic tree generation increases our knowledge on Biosphere evolution, and that is why this topic is actual and original as ever.

The family Thanerocleridae is one of the smallest in Cleroidea and represented by two subfamilies and several genera ranged mainly in America and Tropic Asia and Africa. The most interesting is a fact that this family is represented in the oldest fossil deposition Burmese and Charentees amber. As was mentioned above, all fossil data obtained are useful for phylogenetic study and allows re-construct biota of the past. The present paper provides us with the key to all subfamilies and genera of Thanerocleridae, both living and extinct taxa. This is fundamental result presented for the first time and in combination with new phylogenetic analysis makes paper incomparable with the others by depth of Cleroidea study.

As was declared in the title, the paper is a “Review of the family Thanerocleridae”, and after firsthand acquaintance within we can conclude that it is.

The history of the family and taxonomic structure are considered in the introduction, a new phylogenetic analysis is provided and all taxa known and included to the family are discussed. New method of computed tomography allowed presenting external appearance of the beetle keened in dark and semi-transparent old amber and study small characters necessary to taxonomic analysis, such as type of antenna club, length of tarsomeres, head capsule and palpomeres, puncturation of upperside etc. Of course, this method presents amazing tools for study small fossil animal and plant inclusions and can be recommended to use. Thus, conclusions is argumented by the evidences presented and cover the main idea of the work given in title, the review of the family Thanerocleridae.

Author Response

Thank you for your thorough and positive review. We corrected some technical and formal issues (the holotype information, LSID), formal arrangement of the Key and some minor mistakes. The manuscript was proof-read by a native speaker to work up its English.

Reviewer 3 Report

This paper is beautifully composed and my criticisms are minute.

63. thaneroclerid/-ine
180. the first semicolon is underlined
192. is having --> has
242. wider than segment 8
249. one-third
252. carinated --> carinate
299. delete "the" (last word of the line)
305. one-third
307. arrowhead-shaped

KEY
1. Different characters in the couplets are sometimes separated by commas, other times by semicolons. This should be made consistent throughout the key.
2. Some terminal couplets include the general distribution of the genus. While certainly not necessary, I'd like to see that for all extant genera.
3. America --> USA. "America" could be mistaken as "Americas"

490. Costa Rica (two words)
516. Rio Grande do Sul (full name of the Brazilian State)

Author Response

Thank you for your thorough and positive review. We corrected some technical and formal issues (the holotype information, LSID) and some minor mistakes. All your suggestions were accepted and incorporated (63-516) and the Key rearranged. The manuscript was proof-read by a native speaker to work up its English. Thank you for your help.